# Structure and function of a malaria transmission blocking vaccine targeting Pfs230 and Pfs230-Pfs48/45 proteins

Kavita Singh[1], Martin Burkhardt[2], Sofia Nakuchima[2], Raul Herrera[2], Olga Muratova[2], Apostolos G. Gittis[1], Emily Kelnhofer[2], Karine Reiter[2], Margery Smelkinson[3], Daniel Veltri [4], Bruce J. Swihart[5], Richard Shimp Jr.[2], Vu Nguyen[2], Baoshan Zhang[6], Nicholas J. MacDonald[2], Patrick E. Duffy [2], David N. Garboczi[1] & David L. Narum [2✉]

Proteins Pfs230 and Pfs48/45 are *Plasmodium falciparum* transmission-blocking (TB) vaccine candidates that form a membrane-bound protein complex on gametes. The biological role of Pfs230 or the Pfs230-Pfs48/45 complex remains poorly understood. Here, we present the crystal structure of recombinant Pfs230 domain 1 (Pfs230D1M), a 6-cysteine domain, in complex with the Fab fragment of a TB monoclonal antibody (mAb) 4F12. We observed the arrangement of Pfs230 on the surface of macrogametes differed from that on microgametes, and that Pfs230, with no known membrane anchor, may exist on the membrane surface in the absence of Pfs48/45. 4F12 appears to sterically interfere with Pfs230 function. Combining mAbs against different epitopes of Pfs230D1 or of Pfs230D1 and Pfs48/45, significantly increased TB activity. These studies elucidate a mechanism of action of the Pfs230D1 vaccine, model the functional activity induced by a polyclonal antibody response and support the development of TB vaccines targeting Pfs230D1 and Pfs230D1-Pfs48/45.

[1] Structural Biology Section, Research Technologies Branch, National Institute of Allergy and Infectious Diseases, National Institutes of Health, 29 Lincoln Drive, Bethesda, MD 20892, USA. [2] Laboratory of Malaria Immunology and Vaccinology, National Institute of Allergy and Infectious Diseases, National Institutes of Health, 29 Lincoln Drive, Bethesda, MD 20892, USA. [3] Biological Imaging Section, Research Technologies Branch, National Institute of Allergy and Infectious Diseases, National Institutes of Health, 4 Memorial Drive, Bethesda, MD 20814, USA. [4] Bioinformatics and Computational Biosciences Branch, Office of Cyber Infrastructure and Computational Biology, National Institute of Allergy and Infectious Diseases, National Institutes of Health, 5601 Fishers Lane, Rockville, MD 20852, USA. [5] Biostatistics Research Branch, National Institute of Allergy and Infectious Diseases, National Institutes of Health, 5601 Fishers Lane, Rockville, MD 20852, USA. [6] Vaccine Research Center, National Institute of Allergy and Infectious Diseases, National Institutes of Health, Bethesda, MD 20814, USA. ✉email: dnarum@niaid.nih.gov

A malaria vaccine that targets both human and mosquito infections, also known as a vaccine that interrupts malaria transmission, has the potential to impact malaria elimination and eradication efforts. The importance of developing such a vaccine is underscored by recent WHO World Malaria Reports, which indicate that there has been no significant progress in reducing global malaria cases since 2015[1,2]. The malaria vaccine RTS,S, which targets human infections as assessed by an impact on clinical disease, has completed phase 3 testing[3,4] and is in pilot implementation studies in three African countries[5]. Efforts to develop a vaccine that disrupts mosquito infection, known as a malaria transmission-blocking (TB) vaccine, have been ongoing since the reporting of induced TB immunity in chickens against *Plasmodium gallinaceum* in 1976[6]. The pace of malaria TB vaccine development, until recently, has been hindered by the lack of capacity to produce candidate antigens for clinical testing. With regard to *Plasmodium falciparum*, the field has been limited to clinical testing of Pfs25, a 25 kDa sexual-stage protein present on the *P. falciparum* zygote and ookinete surface in the mosquito midgut[7]. The leading Pfs25 TB vaccine is a chemically conjugated vaccine using a *Pichia pastoris* expressed Pfs25[8] and the carrier protein ExoProtein A (EPA), a recombinant detoxified form of *Pseudomonas aeruginosa* ExoToxin A[9]. The Pfs25-EPA conjugate has the biophysical nature of a nanoparticle with a size similar to the hepatitis B virus-like-particle used in RTS,S[10]. Unfortunately, clinical trial results have not given support to the continued development of a stand-alone Pfs25-EPA TB vaccine. Pfs25-EPA conjugates formulated with Alhydrogel™, an aluminum based adjuvant, in phase 1 trials conducted in the United States[11] and in Mali, West Africa[12] have required four doses to generate antibody titers that significantly reduced parasite transmission as assessed by an ex vivo standard membrane feeding assay (SMFA)[11,12].

Another family of sexual-stage proteins with cysteine-rich domains includes the antigens Pfs230 and Pfs48/45 that have been targeted for TB vaccine development[13–15]. However, until recently, no recombinant form was produced with the identity, purity, and quality necessary for human clinical trials. We reported the first production (to our knowledge) of a cysteine-rich domain from Pfs230, a 230 kDa sexual-stage protein that is composed of fourteen 6-cysteine-rich domains in a manner suitable for human clinical testing[16]. Recombinant Pfs230 domain 1 (Pfs230D1M) was well-characterized and shown to induce TB antibodies in small animals using the SMFA[16]. As part of this work, a conformation-dependent D1-specific TB monoclonal antibody (mAb), identified as 4F12, was produced against parasite-derived Pfs230[16]. Pfs230D1M has been chemically conjugated to EPA following a platform development strategy essentially forming nanoparticles similar in size to hepatitis B virus[10]. A phase 1 safety and immunogenicity study evaluating Pfs230D1-EPA nanoparticles formulated on Alhydrogel™ has been completed (ClinicalTrials.gov ID: NCT02334462) with publication of results currently under peer review. The Pfs230D1-EPA conjugates along with Pfs25-EPA conjugates are currently being evaluated in a more potent liposomal adjuvant AS01 (ClinicalTrials.gov ID: NCT02942277), which is the adjuvant used in RTS,S[4].

Within this family of Pfs230-like proteins, structures have been resolved for two members, which form a complex with an unknown function, Pf12[17,18] and Pf41[19]. Recently, the carboxyl-terminal cysteine-rich domain of Pfs48/45, identified as 6C, was cocrystalized with a TB mAb, which may aid in future vaccine design of a Pfs48/45 immunogen[20,21]. Pfs230 is expressed in both male and female gametocytes without a known membrane anchor and appears on the surface of gametes as a complex with Pfs48/45[22], a glycosylphosphatidylinositol (GPI) anchored protein[23].

Pfs230 is expressed in gametocytes with a prodomain that is processed during gametogenesis and prior to gamete emergence from red blood cells in the mosquito midgut[24]. Studies evaluating Pfs230 function by gene disruption suggest that Pfs230 is necessary for gamete fusion[25]. Here, we developed Pfs230D1-specific mAbs, evaluated the susceptibility of various regions of Pfs230D1 for antibody blockade, and determined the Pfs230D1M protein structure bound to the Fab fragment from the recombinant murine 4F12 antibody. The recombinant 4F12 contained a human IgG1 Fc domain in this study, which has aided in understanding the role of human complement. Using confocal microscopy, we observed that Pfs230 is on the surface of sexual-stage parasites and usually colocalized with Pfs48/45; however, there were clear differences between macrogametes and microgametes. Unexpectedly, some Pfs230 appeared to be on the surface of the macrogamete in the absence of Pfs48/45. Finally, using these unique mAbs along with another Pfs48/45 TB mAb, we assessed the TB function of various mAb combinations. Altogether, we establish the quality of D1 as a vaccine immunogen which would inform, if required, steps to improve future TB vaccines.

## Results

**Structure of Pfs230 domain 1 in complex with the Fab from mAb 4F12.** We previously reported that the mAb 4F12 (Fig. 1), which recognizes a conformational epitope within Pfs230D1, blocked or reduced parasite transmission in the SMFA assay[16]. To produce a recombinant form of mAb 4F12 for

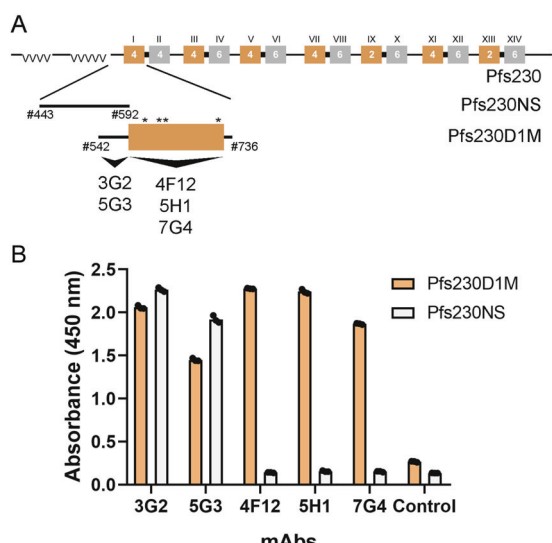

**Fig. 1 Pfs230 domain schematic and mapping of Pfs230D1-specific mAbs. a** Diagram of native Pfs230, two recombinant forms of Pfs230, and the mapping of a new panel of Pfs230D1-specific mAbs. The top line shows the fourteen (Roman numerals) 6-cysteine domains in Pfs230 (orange and grey boxes) that vary in having four or six cysteines. Pfs230D1(I) has four cysteines. The second line shows Pfs230NS (443–592) produced for mAb characterization and mapping (NS nonstructural domain). The third line shows Pfs230D1M (542–736) produced for vaccine trials. Large arrows show the two regions mapped as mAb binding sites. Asterisks (*) represent the relative locations of the four cysteines in Pfs230D1. **b** Epitope mapping of 4F12, previously reported[16], and the new Pfs230D1-specific mAbs by ELISA using the two proteins, Pf230D1M and Pfs230NS. An additional form of Pfs230D1M, identified as Pfs230D1A was produced, which is homologous to Pfs230D1M except for amino acids 585QNT587, which were changed to 585NNA587. See Methods for "Control". Amino-acid residue numbering based on accession number XP_001349600.

**Table 1 Standard membrane feeding assay results for Pfs230D1-specific mAbs.**

| mAb | Isotype | Exp No. | TB activity with human complement | | | | |
|---|---|---|---|---|---|---|---|
| | | | Inf/Diss[a] | Avg No. oocysts | Median No. oocysts (range) | % reduction oocysts | % reduction prevalence |
| 3G2 | IgG2b | 1 | 20/20 | 59.6 | 62 (21–110) | 16.4 | 0 |
| | | 6 | 21/21 | 62.8 | 58 (14–139) | −0.7 | 0 |
| 5G3 | IgG1 | 1 | 20/20 | 68.6 | 66 (20–115) | 3.7 | 0 |
| | | 6 | 21/21 | 47.9 | 39 (8–99) | 23.2 | 0 |
| 5H1 | IgG1 | 1 | 2/20 | 0.1 | 0 (0–1) | 99.9 | 90 |
| | | 2 | 0/20 | 0 | 0 (0–0) | 100 | 100 |
| | | 3 | 15/20 | 1.8 | 1 (0–6) | 93.2 | 25 |
| | | 6 | 21/22 | 4.5 | 4 (0–13) | 92.9 | 4.6 |
| 7G4 | IgG1 | 6 | 21/21 | 74.2 | 68 (6–214) | −19 | 0 |
| | | 7 | 20/20 | 46 | 47.5 (1–83) | 9.3 | 0 |
| 4F12[b] | IgG1 | 1 | 20/20 | 39.4 | 38 (8–76) | 42.2 | 0 |
| | | 2 | 15/20 | 8.4 | 0 (0–38) | 67.1 | 25 |
| 4F12(3B6) | | 4 | 21/21 | 59.6 | 55 (4–118) | 40.3 | 0 |
| | | 6 | 22/22 | 33.8 | 30 (1–64) | 45.9 | 0 |
| rh4F12[b] | mFab/ | 5 | 17/20 | 5.4 | 3.5 (0–27) | 92.2 | 15 |
| | hIgG1 Fc[c] | 8 | 19/20 | 8.1 | 5.5 (0–19) | 84.3 | 5 |

*m* murine, *h* human, *TB* transmission-blocking.
[a]Infected/Dissected.
[b]Percent reduction in oocysts by rh4F12 compared to 4F12 is significantly different ($p = 0.0059$, by Welch Two Sample *t*-test); mAbs were tested at ~1.0 mg/mL IgG except Exp. 8 which was assessed at 0.6 mg/mL.
[c]Positive and negative controls for this assay are shown in Supplementary Table 2.

cocrystallography studies, cDNA was isolated from the 4F12 hybridoma. Unexpectedly, cDNA sequences were observed for two heavy and light chain variable regions; neither were derived from the myeloma fusion partner. The 4F12 hybridoma cells were recloned by limiting dilution and rescreened by ELISA. A subclone was selected, sequenced and identified as 4F12 (3B6). The TB activity of 4F12 and the subclone 4F12(3B6) were comparable (Table 1). A recombinant chimeric 4F12 IgG (rh4F12) molecule was generated using the 4F12 Fab and human IgG1 Fc and shown to reduce transmission by ~two-fold more than the murine 4F12; the increased transmission reducing (TR) activity of rh4F12 was statistically different ($p < 0.0059$, by Welch Two Sample *t*-test) (Table 1). All SMFAs were performed in the presence of human complement.

The complex of Pfs230D1M with the Fab produced from rh4F12 was crystallized and the structure of the complex was determined by molecular replacement (see Methods). This revealed the structure of the Pfs230D1 molecule as a sandwich of two beta-sheets. Pfs230D1 residues 561–730 were visible in the electron density. Not visible were nineteen N-terminal and six C-terminal residues of Pfs230D1 that were present in the recombinant protein and are presumably disordered.

The 4F12 antibody contacts Pfs230D1 along the edges of the two beta-sheets (Fig. 2a). The 4F12 light chain makes 98 contacts between atoms of 4F12 and atoms of Pfs230D1 that are less than four angstroms apart. The heavy chain makes 12 such contacts to Pfs230D1. The light chain contacts Pfs230D1 residues K581, Y582, A583, S584, Q585, and N586 in the first beta-strand and residue D594 in the second beta-strand in the sequence. There are light chain contacts to the loop between the second and third beta-strands at T596, D597, Q598, K600, T602, E603, S604, and K607. The light chain also contacts the third beta-strand at K609. The first disulfide (593–611) links the two beta-sheets at the second and third strands and appears to stabilize the 4F12 epitope on Pfs230D1. The 4F12 heavy chain binds Pfs230D1 at the loop between the second and third beta-strands of Pfs230D1, contacting P601 and overlapping the light chain contacts to atoms of D597 and K600.

The production of recombinant proteins in *P. pastoris* runs the risk of NxT/S N-linked sites being glycosylated, when such sites are not glycosylated in *Plasmodium*. There is one predicted N-linked site in the Pfs230D1 sequence, which would direct the addition of carbohydrate to N585. N585 was changed to a glutamine to avoid possible glycosylation in the production of recombinant Pfs230D1M[16].

The mAb 4F12 was generated by immunizing with and then screening hybridomas against *P. falciparum* gametocyte preparations, which would have carried N585, the asparagine of the NxT/S predicted site. Position 585 is within the binding site of 4F12 in the structure of the complex with Pfs230D1M. The affinity of binding in vitro between 4F12 and Pfs230D1M was measured to be 23.8 nM by isothermal titration calorimetry (Supplementary Fig. 1A). The nature of this binding was further evaluated using an alternative recombinant Pfs230D1 in which the N585 was restored and T587 was mutated to A587, which is identified as Pfs230D1A. Purified Pfs230D1A appeared highly comparable to Pfs230D1M. The full biochemical and biophysical characterizations of Pfs230D1A are reported in Supplementary Fig. 2. The affinity of binding in vitro between 4F12 and Pfs230D1A by isothermal titration calorimetry (Supplementary Fig. 1B) was highly similar to Pfs230D1M, measured to be 45.2 nM. Despite the Q at position 585, the antibody exhibits a high affinity for Pfs230D1M.

**Comparison with known 6-cysteine-rich structures**. The structure of Pfs230D1 resembles those of other members of the 6-cysteine family. The shape and make-up of Pfs230D1 has similarity with the domains of Pf12, Pf41, and Pf48/45 of the 6-cysteine family (Fig. 2b), although Pfs230D1 has four cysteines instead of six. The two disulfide bonds of Pfs230D1 are Cys593-Cys611 (C1-C2) and Cys626-Cys706 (C3-C6) confirming those first determined in MacDonald et al.[16]. The two disulfides closely overlay the analogous bonds in the other 6-cysteine domains. The disulfide that is not present in Pfs230D1 would be identified as C4-C5 in the 6-cysteine structures. The analogous residues for the C4-C5 disulfide in Pfs230D1 are Ser646 and Phe704.

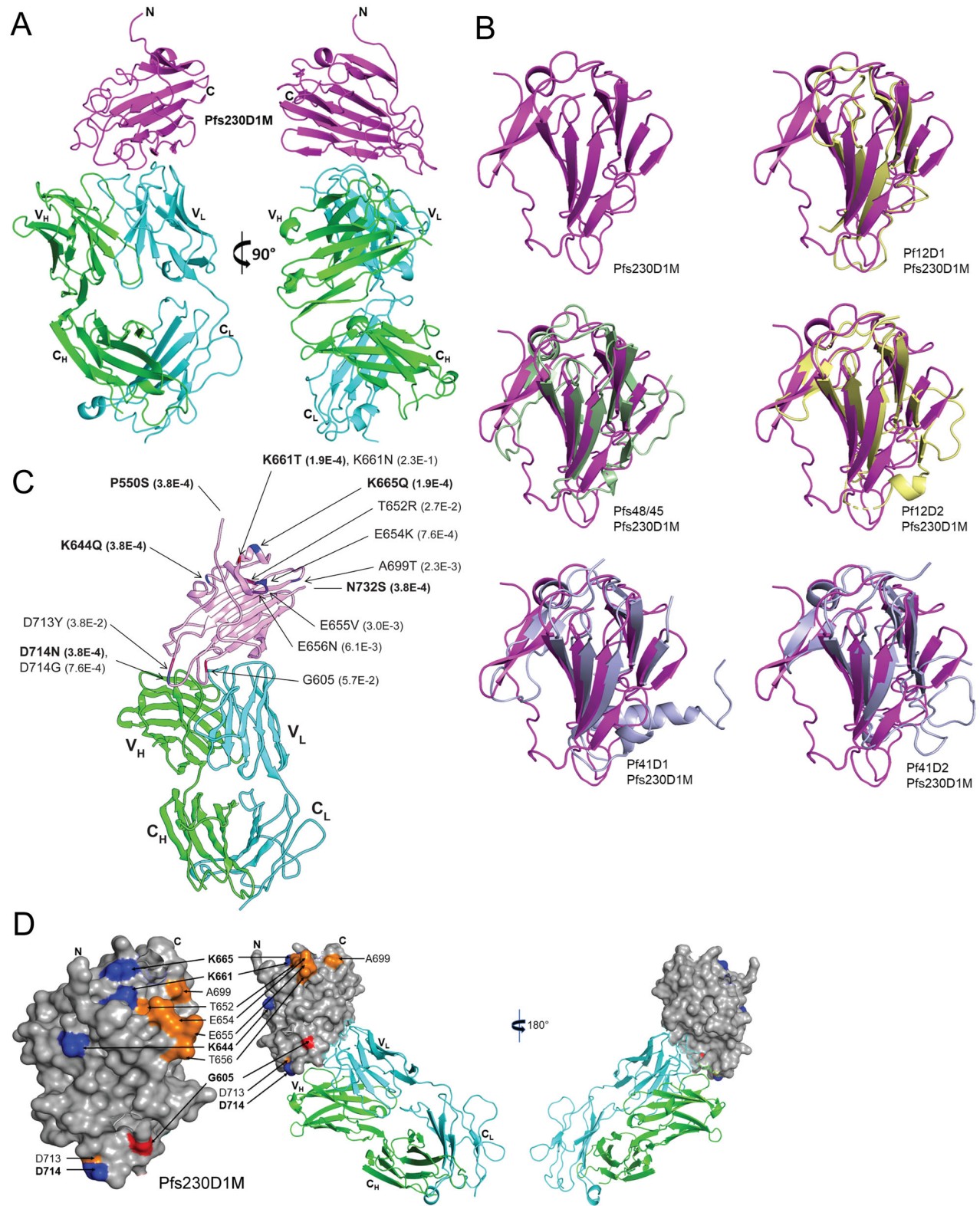

Both domains of Pf12 (PDB code: 2YMO) overlay closely on Pfs230D1. The first domain of Pf12 superposes on Pfs230D1 with a root mean square deviation (rmsd) of 1.9Å over 82 α-carbons and the second domain with an rmsd of 2.4Å over 86 α-carbons. Both domains of Pf41 (PDB code: 4YS4) overlay well on Pfs230D1. Domain 1 of Pf41 matches the structure of Pfs230D1 with an rmsd of 1.7Å over 100 α-carbons and domain 2 of Pf41 with an rmsd of 2.1Å over 102 α-carbons. The structure of the

Pf48/45 C-terminal domain (PDB code: 6E62) overlays Pfs230D1 with an rmsd of 2.2Å over 91 α-carbons.

In each of the five superpositions in Fig. 2b, the beta-stranded cores of the molecules can be seen to be similar and overlay each other well. Interestingly, at the end of each domain nearer to the C-terminus, the loops that connect the strands are also similar in length. By comparison, at the other end of the molecules closer to the N-termini of the domains, the loops of all six domains are

**Fig. 2 Analysis of Pfs230D1M-Fab structure and polymorphisms. a** Two views of the structure of Pfs230D1 complexed with the 4F12 Fab. Pfs230D1 (magenta) shown as a ribbon diagram bound to the light (cyan) and heavy (green) chains. The views are separated by 90 degrees around the vertical axis. VL, VH, light or heavy chain variable regions. CL, CH light or heavy chain constant domains. N N-terminus. C C-terminus. **b** Superpositions of 6-cys family members on Pfs230D1. Upper left: Ribbon diagram of Pfs230D1 (magenta) alone. Upper right: Pf12 domain 1 (yellow) on Pfs230D1. Middle left: Pf48/45 domain 3 (green) overlaid on Pfs230D1. Middle right: Pf12 domain 2 (yellow) on Pfs230D1. Lower left: Pf41 domain 1 (gray) on Pfs230D1. Lower right: Pf41 domain 2 (gray) on Pfs230D1. The orientation of Pfs230D1 is identical in all panels. The "C-terminal" ends of the domains where both beta-strands and loops overlay closely are pointing down, at the "bottom" of the domains. The "N-terminal" ends where the loops diverge are at the "top" of the domains. **c** Location and minor allele frequency (MAF) of the thirteen sequence variants within the crystal structure (lines with arrows) and two variants just outside the structure (lines without arrows). The reference amino-acid G605 identified in NF54 is labelled without a mutation as it was determined to be the minor allele. Bolded residues of low frequency variants (MAF < 0.001) were those with the highest shared occurrence rate. The bolded resides would be likely to appear in the same samples more frequently than others. See text and methods. **d** Left, surface representation of the Pfs230D1 polymorphisms mapped on to the Pfs230D1 structure. The residues with the highest colocalized occurrence rate are bolded and their surfaces are in blue. The other polymorphisms are denoted in orange on the figure. The residue G605 is in red. For clarity, the labels for A699, D713, and D714 are given twice. Right, Pfs230D1 in surface representation (gray) and Fab 4F12 in ribbon, showing that 4F12 binds to a surface of Pfs230D1 that is not polymorphic. The closest polymorphic residue to the epitope is G605 (red), which does not contact the Fab.

much more divergent in their lengths and positions, and thus do not overlay. From these six 6-cysteine domains, we generalize that one end of the beta-sandwich of a 6-cysteine domain will have shorter loops that closely overlay, and that the other end of the domain will have longer, divergent loops connecting the beta-strands.

**Analysis of amino-acid polymorphism within Pfs230D1M boundaries.** We previously determined the polymorphic residues within Pfs230D1M and identified two predominant amino-acid substitutions G605S and K661N[16]. We observed an apparent colocalization of these major amino-acid changes on a part of the structure located distal to the 4F12 epitope. We furthered this analysis using the latest variant data from the MalariaGEN Pf3k project pilot release 5 (https://www.malariagen.net/projects/Pf3k) and assessed whether this colocalization was by chance or non-randomly associated across 2618 *P. falciparum* samples. As the Pf3k dataset consists of de novo calls, these should be considered as candidate variants until further studies can confirm any biological effects.

Fifteen single nucleotide variants predicted to cause missense mutations at thirteen amino-acid residue positions were identified and are labeled with accompanying allele frequencies (AF) on the Pfs230D1 structure shown in Fig. 2c. As the determined structure includes sequence positions 561 through 730, we note variants P550S and N732S point to the N- and C-termini, respectively, as their amino-acid residues occur just outside this range but within D1 of Pfs230 (Fig. 2c). As described in Methods, the missense variant G605S was identified but regarded as the major allele due to its high AF of 0.94. This variant is listed under ID 31823 in the Pf3k release 3 web application (https://www.malariagen.net/apps/pf3k/release_3/index.html) but is not available for the release 5 data at the time of this writing. For consistency, we use the minor (reference) allele G605 for labeling in Fig. 2c and the analyses described in Supplementary Fig. 2. Amongst the labeled variants, five occur within the same beta-turn, one within a small alpha-helix, and the rest within coiled regions. Labels for variants P550S, K644Q, K661T, K665Q, D714N, and N732S (Fig. 2c) are marked in bold to denote their rarity (minor AF < 0.001). These variants were found to have the highest colocalized occurrence rate across all samples based on pairwise Pearson moment correlation ($r = 0.67$, Benjamini and Hochberg corrected $p$-values < 0.001 for each pair) as further described in Methods. These same rare variants were also seen to cluster (Supplementary Fig. 3A)—forming an aqua green hotspot in the lower left of the Pearson's $r$ value heatmap, as well as a distinct cluster near the origin of the t-Stochastic Neighbor Embedding (t-SNE) plot (Supplementary Fig. 1B) The t-SNE plot is a two-dimensional projection of binary variant presence/absence data for all 2618 samples. Accordingly, amino-acid substitution labels that are closer in proximity have distributions of minor alleles that are more similar across samples. We also noted that K661N and T652R form a cluster near the top right of the heatmap ($r = 0.28$, $p < 0.001$) and appear nearby on the t-SNE plot as well.

**Development of additional Pfs230D1-specific mAbs that have TB activity.** To better characterize Pfs230D1 we developed an expanded mAb panel that recognized Pfs230D1M. Following the initial screen, four mAbs were selected for expansion and characterization. The four mAbs were shown to recognize two distinct regions of Pfs230D1 by ELISA using two different recombinant proteins that provided coverage of distinct regions of the amino-terminus of Pfs230 through D1 (Fig. 1b and Supplementary Fig. 4). By ELISA and western blot analysis, mAbs 3G2 and 5G3 were observed to react with a linear epitope within the non-structured amino-terminal end (residues 542–592), while the second set, 5H1 and 7G4, mapped to the disulfide constrained region of D1 and both were reduction sensitive (Supplementary Fig. 4). Only one of the new mAbs (i.e., 5H1, Table 1) was shown to provide transmission reducing (TR) activity using the SMFA and yielded a significant reduction in the prevalence of a mosquito infection. The other three Pfs230D1 mAbs had no demonstrable TR activity (Table 1). The TB activity of 5H1 was shown to be titratable based on IgG concentration as well (Supplementary Fig. 5A). It has not yet been possible to clone and express a recombinant form of 5H1. Human complement was included in all assays. To assure assay performance, two control groups were included in the assay. The positive control used extensively in this assay[26] was mAb 4B7, which targets the zygote and ookinete surface protein Pfs25[27], and the negative control was a mAb against a *P. yoelii* asexual protein (Supplementary Table 2)[28,29].

**Distinct pattern of recognition on live sexual-stage parasites by Pfs230D1 mAb panel.** The surface accessibility of the epitopes recognized by the Pfs230D1 mAb panel was investigated. We performed live labeling studies such that live macrogametes were labeled with each mAb then, following fixation, counter-stained with the Pfs230D1-specific polyclonal antibodies (pAbs) as a control to ensure Pfs230 expression and to avoid the potential binding interference of the mAbs. We observed that only mAb 4F12 recognized live sexual-stage parasites, specifically macro-gametes (Fig. 3a). To confirm that the mAbs could recognize Pfs230D1, we determined their reactivity to Pfs230 on fixed parasites and demonstrated that macrogametes labeled live with

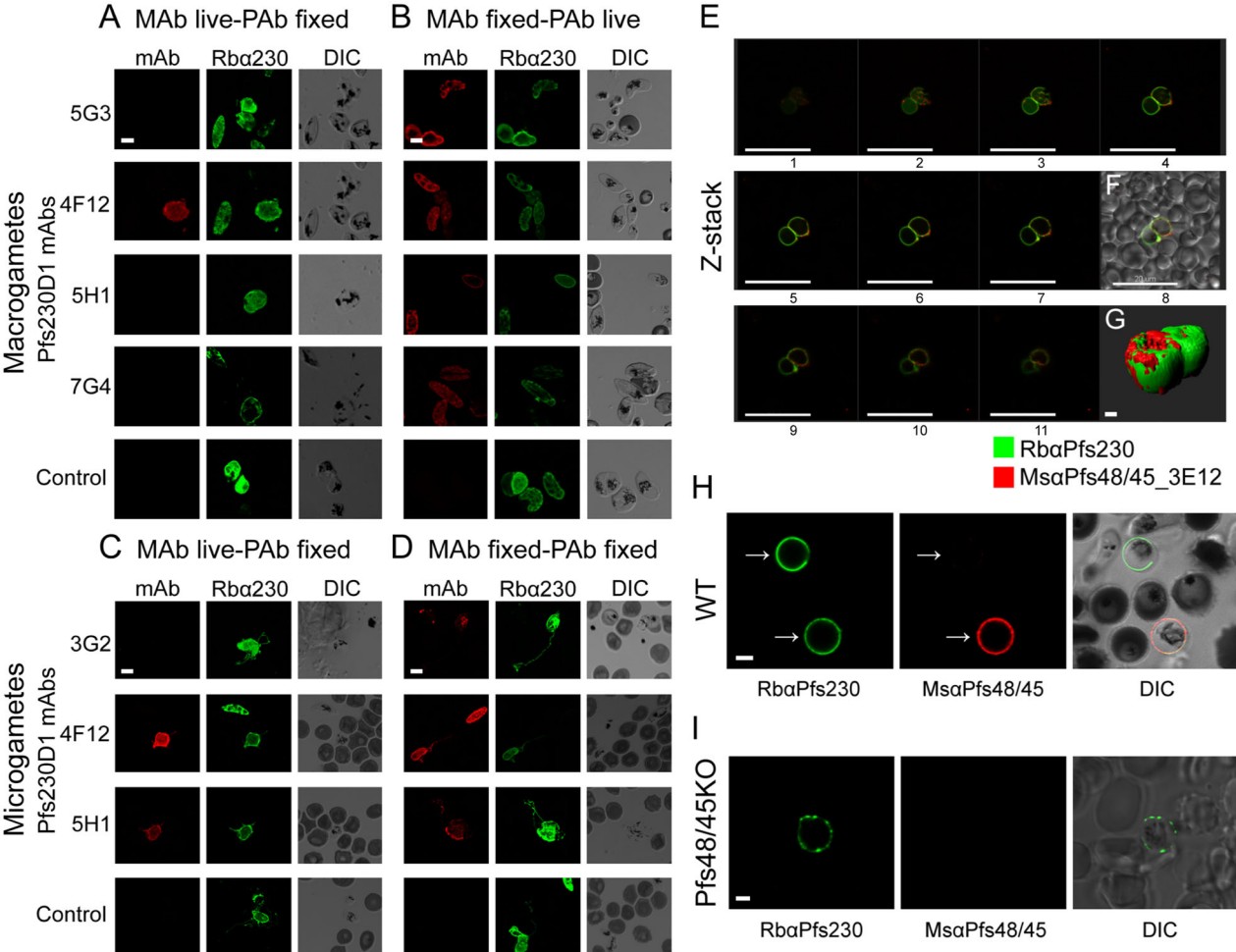

**Fig. 3 Pfs230, Pfs48/45, and Pfs230-Pfs48/45 subcellular localization in *P. falciparum* macrogametes and microgametes by confocal microscopy under live and/or fixed conditions. a** Live emerged macrogametes were labeled with Pfs230D1-specific mAbs (*mAb*) as listed at the left. To demonstrate the presence of Pfs230, macrogametes were fixed and labeled with Pfs230D1-specific rabbit polyclonal (p) Abs (*Rbα230*). The third column shows the same cells by differential interference contrast (*DIC*) microscopy. 4F12 reacted with the surfaces of live macrogametes, but that 5G3, 5H1, and 7G4 did not. **b** Live emerged macrogametes were labeled with pAbs (*Rbα230*) then fixed and stained with the same Pfs230D1-specific mAbs as panel A. After fixation all mAbs reacted with the surface of the macrogametes. **c** Live microgametes were labeled with Pfs230D1-specific mAbs as listed at left. To demonstrate the presence of Pfs230, they were fixed and labeled with Pfs230D1-specific pAbs (*Rbα230*). 4F12 and 5H1 recognized live microgametes, unlike 3G2. **d** Fixed microgametes were labeled with pAbs (*Rbα230*) and with Pfs230D1-specific mAbs. After fixation, all mAbs recognize Pfs230. Only 4F12 recognized live macrogametes in A while both 4F12 and 5H1 recognized live microgametes in **c**. **e** Pfs230 and Pfs48/45 colocalize on the macrogamete surface membrane with live staining using the Pfs230-specific pAbs (*RbαPfs230*, green) and mAb 3E12 (*MsαPfs48/45_3E12*, red). **f** inset showing DIC of **e**. **g** Three-dimensional surface distribution of the Pfs230 and Pfs48/45 proteins. **h** Staining of wild-type (*WT*) NF54 emerged macrogametes labelled live with rabbit Pfs230D1-specific pAbs (green, *RbαPfs230*) and mouse Pfs48/45 domain 3 pAbs (red, *MsαPfs48/45*) show distinct staining patterns. At left, two macrogametes stain with Pfs230-specific mAb 4F12 (arrows). In the middle, only the lower of the two macrogametes is recognized by Pfs48/45-specific pAbs (arrows). At right, DIC shows the overlay of both stainings. **i** Pfs230 appears to be membrane associated without Pfs48/45. Live NF54 Pfs48/45 knockout (KO) macrogamete labels with rabbit Pfs230D1-specific pAbs (green, *RbαPfs230*), but is not labelled with mouse Pfs48/45 domain 3 pAbs (red, *MsαPfs48/45*). DIC shows both Abs overlaid. Scale bars in **a–d** and **g** represent 2 μm, **h–i** represent 3 μm, while in **e** and **f** represent 20 μm.

rabbit Pfs230D1-specific IgG were counter-stained with each mAb following fixation (Fig. 3b). We repeated these studies using exflagellating sexual-stage parasites in order to live-label microgametes (see Methods). By live labeling, we observed that only 5H1 in addition to 4F12 bound microgametes while 3G2 did not (Fig. 3c). As expected, we observed microgamete recognition by the entire mAb panel following fixation and the staining colocalized with Pfs230D1 pAbs (Fig. 3d).

**Pfs230 appears membrane associated in the absence of Pfs48/ 45.** Pfs230 has no apparent membrane anchor and is known to form a complex with Pfs48/45 which contains a GPI anchor[22]. Given this protein–protein association, we studied the Pfs230-

Pfs48/45 complex on the surface of live macrogametes using Pfs230D1-specific reagents in association with a TB Pfs48/45-specific mAb 3E12[30] or Pfs48/45 pAbs generated against different recombinant forms of Pfs48/45 (Supplementary Fig. 6A–C)[30] on macrogametes. During this study, we mapped 3E12 to domain 3 of Pfs48/45 (Supplementary Fig. 6D). We observed co-staining on the surface of macrogametes, which predominately colocalized when individual parasites were captured on poly-lysine-treated plates (Fig. 3e, f). Furthermore, this analysis allowed for a 3D rendering of the Pfs230 and Pfs48/45 molecules on the macrogamete (Fig. 3e inset labeled G).

Unexpectedly, we also observed the presence of Pfs230D1-specific IgG on the surface of live macrogametes in the absence of

Pfs48/45 mAb 3E12 (Fig. 3h). We quantified the frequency of this occurrence using an automated process on more than 2000 live parasites co-stained for Pfs230D1 and Pfs48/45. We observed ~11.9% (259/2160) of the macrogametes staining for Pfs230 only on the membrane surface without Pfs48/45, 34% (734/2160) were stained for both Pfs230 and Pfs48/45, and 54.0% (1167/2160) were Pfs48/45 only. We hypothesized that the unexpectedly large proportion of Pfs48/45 single positives was due to a loss of Pfs230 possibly due to its release in the media or degradation during the time of incubation (~4 h). This was confirmed by co-staining fixed sexual-stage parasite thin films with Pfs230 and Pfs48/45, then observing that 96% (96/100) were double positive for Pfs230 and Pfs48/45. Finally, we confirmed the observation that Pfs230 can be membrane associated in the absence of Pfs48/45 by detecting Pfs230 on live-labeled macrogametes from a Pfs48/45 knockout (KO) NF54 parasite line (Fig. 3i and Supplementary Fig. 6F).

**Combinations of Pfs230D1 and Pfs48/45-specific mAbs enhance TB activity.** Given that 4F12 and 5H1 both had TB activity, we investigated whether they recognized a similar or different epitope. Using a competition ELISA assay, we observed that 4F12 and 5H1 did not compete with each other for binding and therefore did not recognize the same epitope (Fig. 4a). Since each mAb recognized a distinct epitope, we evaluated whether combining the two mAbs would provide an increase in TR activity compared to each alone. We assessed the activity of both 4F12 or rh4F12 in combination with 5H1 at concentrations below those that completely block transmission and found that the combinations demonstrated enhanced TR activity (Fig. 4b, c). These observations were confirmed in independent studies (Supplementary Fig. 5B, C). A statistical analysis using zero-inflated negative binomial models confirmed that these observable differences were significant ($p < 0.001$).

Next, we investigated the potential benefit of combining mAbs specific to Pfs230D1 and Pfs48/45 to represent the coadministration of a combination vaccine. To allow for better control in the SMFA, we selected Pfs230D1-specific mAb 5H1 and Pfs48/45-specific mAb 3E12, using each alone and in combination. Prior to initiating this study, we established that the TR activity of mAb 3E12 was titratable (Supplementary Fig. 5A). The results of the combination, shown in Fig. 4d (and Supplementary Fig. 5D), demonstrate a significant increase in TR activity compared to each mAb alone ($p < 0.001$). In contrast, we also assessed whether a combination of Pfs230D1 and Pfs25 mAbs or pAbs would increase the TR activity, but no enhancement was observed (Supplementary Fig. 5E, F).

## Discussion

The development of a malaria TB vaccine has the potential to augment existing control strategies and enable an assessment of a combination vaccine targeted to interrupt malaria transmission to both the human and mosquito host. Until a few years ago, the only *P. falciparum* TB vaccine candidate to be evaluated in human trials was Pfs25[11,12]. Since developing a scalable platform for producing Pfs230D1[16] following current good manufacturing practices (cGMP), Pfs230D1M has entered clinical testing as an alternative candidate or combination with Pfs25 (clinicaltrials.gov ID NCT02334462). In this work, we resolved the structure of domain 1 from Pfs230 in complex with the conformation-dependent mAb, 4F12, raised against NF54 sexual-stage parasites.

Pfs230 is one of the largest members of the 6-cysteine family. The 4F12 mAb binds one side of Pfs230D1 and buries an average amount of surface area for antigen-antibody complexes of about 1500 Å$^2$. The beta-sheet sandwich of the domain was revealed to

be similar to domains of other 6-cysteine proteins. The six 6-cysteine domains with available structures overlay structurally with an rmsd of about 2 Å. We observed that one end of all six domains has short loops and the other end of each domain has long loops. These observed differences in the loops of the known domains may stem from the requirements of packing 6-cysteine domains in multi-domain proteins.

The Pfs230D1-specific mAb 4F12 reduces transmission in the presence or absence of human complement[16] especially since it is generally accepted that murine IgG1 does not fix human complement. Consistent with earlier findings regarding the role of antibody isotype and TB activity[31,32], when the Fc region of 4F12 was replaced with a human IgG1 Fc domain, which may fix complement, the TR activity increased significantly (Table 1, $p = 0.0059$, by Welch Two Sample $t$-test). These findings indicate that antibodies against Pfs230D1 sterically interfere with a protein–protein interaction independent of their potential to activate human complement and lead to membrane lysis. Eksi et al.[25] reported for *P. falciparum* that an N-terminal truncation of Pfs230 altered the phenotype of microgametes to adhere to uninfected RBCs. Thus, whether 4F12 interferes with a parasite-host interaction (e.g., RBC binding) or a parasite-parasite interaction is unknown. What has been reported previously is an association with Pfs230-specific mAbs[31] or pAbs[33] with TB activity due to human complement.

Polymorphism within a vaccine candidate is an important consideration to assess during development. A malaria vaccine candidate targeting clinical disease was initially viewed to have limited variation[34] but this assessment has changed over time as more sequences were published[35]. Our continued evaluation of the missense coding changes or amino-acid polymorphisms within the boundaries of Pfs230D1 suggest that there is some form of selective pressure on D1. Of the 14 candidate missense coding changes highlighted in Fig. 2c, correlation analysis identified a cluster of six rare variants (P550S, K644Q, K661T, K665Q, D714N, and N732S) with higher rates of profiles across *P. falciparum* samples (Supplementary Fig. 3A). It is interesting that all six of these coding changes result in polar uncharged amino acids and these may serve as targets of interest for further evolutionary analysis of Pfs230D1. However, as they are based on de novo variant call data, they first require validation and further assessment of impacts on protein structure and function. It is unclear whether this selective pressure is due to the low level of natural human antibodies present against Pfs230[36], and in particular D1, or pressures related to the mosquito microenvironment and sexual development during transmission[37]. We previously observed in West Africa that the most abundant base changes had a minor allelic frequency of 0.111 and 0.339 for G605S (identified in this work as G605) and K661N, respectively, which did not impact the TR activity of rabbit Pfs230D1-specific IgG in a membrane feeding assay[16]. In this analysis, we observed a statistically significant correlation between K661N and T652R even though the minor allelic frequency of T652R is small (0.027). It will be important to continue to monitor missense coding changes in wild-type parasites during ongoing field trials.

The importance of structure-based vaccine design has been well established for respiratory syncytial virus[38,39]. The implications for future design of D1, if any, will need to be carefully considered. The Pfs230D1M cGMP production in *P. pastoris* is scalable, efficient and uses a cost-effective system[40]. Furthermore, it is known that at least two functional epitopes are available, consistent with the functional activity of a polyclonal antibody response. It was surprising that the N-linked glycosylation site mutation (N585Q) introduced in recombinant Pfs230D1M was within the 4F12 binding site. The formation of complex N-linked sugars at consensus sites (NxT/S) in *Plasmodium* appears

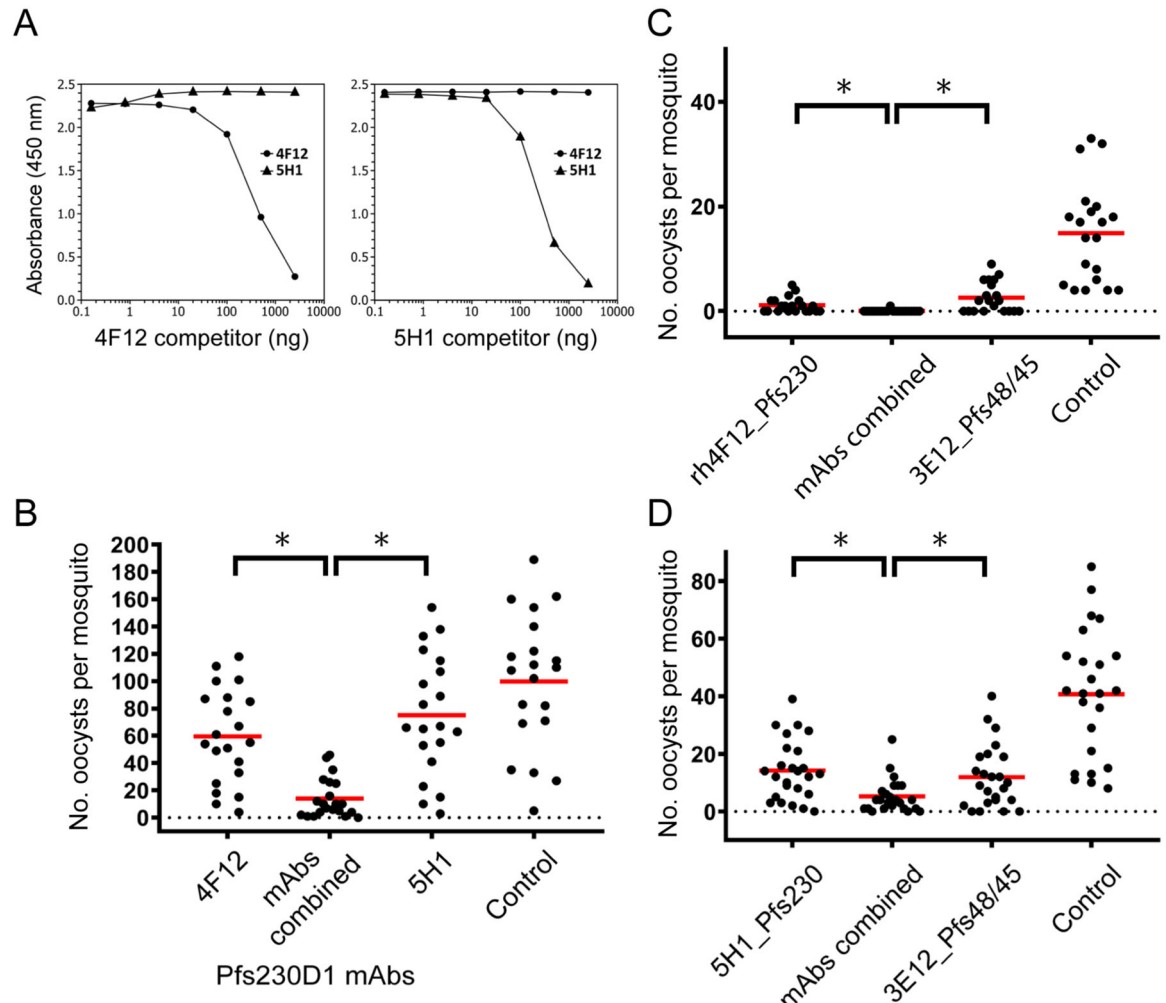

**Fig. 4 Binding competition by ELISA of Pfs230D1-specific mAbs and the functional activities of Pfs230D1 and Pfs48/45-specific mAbs in an ex vivo feeding assay. a** In both panels, 4F12 (circles) and 5H1 (triangles) were bound to ELISA plates. On the left, Pfs230D1-specific mAb 4F12 was added as competitor, resulting in competition on the 4F12 plate (circles), but no competition on the 5H1 plate (triangles). On the right, is the analogous experiment, but with 5H1 as the competitor. 4F12 and 5H1 do not compete for binding to Pfs230D1 as seen by competitive ELISA. As a control in each panel, it can be seen that each mAb does compete against itself. **b** Combining Pfs230D1-specific mAbs, 4F12 and 5H1, show enhanced TR activity ("mAbs combined") compared to each set of mAbs alone (*$p < 0.001$). The average inhibition of oocyst density was 40.3%, 86.0%, and 24.8% against the control (0%) at IgG concentrations: 1.1, 1.1 + 0.63, 0.63, and 1.0 mg/mL, respectively. **c** Combining Pfs230D1-specific mAb rh4F12 ("rh4F12_Pfs230") and Pfs48/45-specific mAb 3E12 ("3E12_Pfs48/45") shows enhanced TB activity ("mAbs combined") compared to each set of mAbs alone (*$p < 0.001$). The average inhibition of oocyst density was 92.3%, 99.4%, and 82.6% against the control (0%) at IgG concentrations: 0.63, 0.63 + 0.04, 0.04, and 0.94 mg/mL, respectively. **d** Combining Pfs230D1-specific mAb 5H1 ("5H1_Pfs230") and Pfs48/45-specific mAb 3E12 ("3E12_Pfs48/45") shows enhanced TB activity ("mAbs combined") compared to each set of mAbs alone (*$p < 0.001$). The average inhibition of oocyst density was 65.1%, 87.1%, and 70.8% against the control mAb 48F8 (0%) at IgG concentrations: 0.25, 0.25 + 0.04, 0.04, and 0.35 mg/mL, respectively.

limited[41,42] while *P. pastoris* may generate complex N-linked glycosylated proteins[43]. N-linked glycosylation of recombinant malaria protein vaccine candidates may negatively impact immunogenicity[44] or protein quality[45]. A further benefit to these mutations is improved recombinant protein productivity due to an increased homogeneity while in general, not observing any biological impact on induction of polyclonal antibodies or their function[46,47]. Here, we determined that the mutation introduced in Pfs230D1 is within the 4F12 epitope and has apparently created limited structural changes based on the 4F12 binding constant of 23.8 nM, which is further supported by our analysis of Pfs230D1A which showed a similar binding constant. The introduction of the N585 mutation appears to be a neutral change to the vaccine immunogen.

Pfs230 domain 1 is an effective immunogen and was recently reconfirmed as the most robust region for antibody targeting based on production of antibodies against numerous fragments covering Pfs230, a 230 kDa protein[32]. The short N-terminal nonstructured region contained in Pfs230D1 appears to be largely unexposed in sexual-stage parasites, and as a result mAbs that recognize this region are not likely to interfere with transmission. We saw a similar absence of parasite recognition by mAb 7G4 and no TR activity (Table 1). These results indicate that D1 is packed within the larger Pfs230 protein such that only select surfaces are exposed to the aqueous environment. The observed differences between surface recognition by 4F12 and 5H1 on microgametes versus 4F12 only on macrogametes indicate that Pfs230D1 epitopes are presented differently on sexual-stage parasite surfaces. Understanding the domain packing of Pfs230D1 based on epitope exposure appears even more complicated given the observation that Pfs230 may be membrane associated in the absence of its known binding partner Pfs48/45, a

gpi anchored protein[22,23]. The biological basis for the membrane association of Pfs230 in the absence of Pfs48/45 remains unclear. Pfs230 is a large 230 kDa protein with fourteen 6-cysteine-rich domains. If other domain(s) contribute to its membrane association or potential to bind to other membrane-associated proteins remains to be determined. We have not identified any novel sexual-stage proteins following immunoprecipitation of Pfs230 from solubilized NF54 sexual-stage parasites other than Pfs48/45 by tryptic digestion followed by liquid chromatography with tandem mass spectrometry. Additional work is warranted using the Pfs48/45KO parasite line.

If Pfs230D1 alone is insufficient as a TB vaccine, a combination TB vaccine that includes Pfs48/45 may be a stronger candidate. Recently, a Pfs230 and Pfs48/45 fusion protein comprised of a fragment of the Pfs230 prodomain which is upstream of domain 1, similar to Pfs230NS, and Pfs48/45 6C domain (domain 3) was shown to induce a three-fold increase in functional titers in a complement independent manner[48]. The enhanced biological activity may have been due to an increased antibody titer against Pfs48/45 6C since it was reported that a recombinant form of the Pfs230 prodomain alone did not induce TR antibodies[32]. Clearly, a combination vaccine composed of Pfs230D1 and Pfs48/45D3 has the potential to improve the robustness of both TB and durability due to the functional activity of each specific polyclonal antibody acting at lower concentrations.

The development of Pfs230D1 as a conjugate vaccine to form a nanoparticle[49], has provided a significant step forward toward development of a malaria TB vaccine. If the result from the current Phase 2 trial of a Pfs230D1-EPA conjugated nanoparticle formulated in AS01 (clinicaltrials.gov ID NCT03917654) shows the vaccine to be insufficient, then a clear next step is further investment in the development of Pfs48/45 using a scalable process with good productivity and quality for inclusion of Pfs48/45 as an additional TB vaccine component. Pfs230D1 should remain as a base component of a TB vaccine given that antibody acts directly against sexual-stage parasite development and that TB activity is enhanced with human complement.

## Methods

**Recombinant protein production.** The production of Pfs230D1M has been previously reported and the *P. pastoris* Pfs230D1A protein was prepared similarly to Pfs230D1M[16]. The nucleotides CAG AAT ACC of Pfs230D1M encoding residues QNT were changed to AAC AAT GCC encoding residues NNA in the new Pfs230D1A, which maintained the disruption of the N-linked glycosylation site at the wild-type residues NNT. The *E. coli* expressed Pfs230NS (XP_001349600.1 Glu443 to Val592) and Pfs48/45 domain 3 (Pfs48/45D3C) proteins were encoded by genes codon optimized for expression in *E. coli*. The synthetic genes were cloned into the NdeI and XhoI sites of pET-24a(+) (Novagen) downstream of the T7 promoter. The expression plasmids were transformed into the *E. coli* BL21(DE3) (Invitrogen) or T7 Express (NEB) competent cells for recombinant expression. The *E. coli* expressed Pfs230NS and Pfs48/45 domain 3 proteins contained heterologous LEHHHHHH amino acids at their carboxyl terminus. Pfs48/45 domain 1–2 was expressed in *P. pastoris* GS115 using a codon optimized synthetic gene (BioNexus, CA) encoding the ectodomains 1 to 2 (Genbank accession number Z22145, Asp28 to Lys349) cloned in the pPIC9K expression vector following the manufacturer's guidelines (Thermo Fisher Scientific). Three putative N-linked glycosylation sites were mutated as follows: S52A and T133A and S206A. The Pfs48/45D1-2 research clone was subsequently transformed with *P. pastoris* protein disulfide isomerase as previously described[8]. Recombinant Pfs48/45D1-2 protein contained the heterologous HHHHHH amino acids at the carboxyl terminus.

The *E. coli* Pfs48/45D3 cells were grown in LB broth. Inclusion body preparations were refolded by dilution and purified using Toyopearl MX-Trp-605M (Tosoh Biosciences), a mixed mode resin and then size exclusion chromatography using a Superdex 75 1.6 × 60 cm column (GE Healthcare) equilibrated with PBS, pH 7.4 following procedures developed previously[50]. The *P. pastoris* Pfs48/45D1-2 clone was fermented in 5 L Eppendorf fermenters using defined media as previously described and used pH 4.0 and 25 °C during induction. The fermentation supernatant was recovered by centrifugation, dialyzed into 2X PBS, pH 7.4 and processed through a Nickel Sepharose (GE Healthcare) affinity column and eluted with imidazole/PBS solution. The purified recombinant Pfs48/45D1-2 was then concentrated and further purified by size exclusion

chromatography (1 × 30 cm Superdex 75 column (GE Healthcare). The elution peaks for each product were pooled and stored at −80 °C.

**Generation of 6-cysteine-rich domain-specific antisera.** Female BALB/c mice, 6–8 weeks of age or female New Zealand White rabbits, 12–14 weeks were immunized thrice with purified recombinant protein formulated in a water-in-oil adjuvant i.e., Freund's or Montanide ISA720 following a 14- or 21-day schedule as previously described[16]. Sera were collected 14 or 21 days following the second boost and used for the assays described here or Protein G affinity purified following procedures as previously described[16]. All animals used for this project were approved by the National Institute of Allergy and Infectious Diseases, Division of Intramural Research, Animal Care and Use Committee, protocol: ASP LMIV 1E at the National Institutes of Health, which is AAALAC accredited and OLAW assured. The NIAID DIR Animal Care and Use Program acknowledges and accepts responsibility for the care and use of animals involved in activities covered by the NIH IRP's PHS Assurance #A4149-01.

**Monoclonal antibody and recombinant antibody production.** Pfs230D1-specific mAbs were generated by Precision Antibody™ (Columbia, MD) using EcPfs230D1-2 or Pfs230D1M as immunogens[16], as previously reported[50]. Select hybridomas secreting IgG were expanded in vitro and culture supernatants were used for Protein G purification as recommended by the manufacturer (Pierce/Thermo Fisher Scientific). The mAb isotyping was performed as previously described[50]. Nucleotide sequencing of mAb 4F12 was performed by LakePharma, CA. The accession numbers for the 4F12 H and L chain are MK756320 and MK756321, respectively.

Heavy and light chains from the 4F12 hybridoma were amplified and ligated into the pVRC8400 vector as previously described[51]. One liter of HEK293 FreeStyle cells (Invitrogen catalog number R79007) at $1 \times 10^6$ cells/ml in FreeStyle media (Invitrogen catalog number 12338021) were simultaneously transfected with 0.5 mg of heavy and light chain vector with 1 mL 293fectin (Invitrogen catalog number 12347019) according to the manufacturers protocol to produce recombinant antibody. After 120 h, supernatant containing antibody was loaded onto a 5 ml protein A column. Following 20 column volumes of wash with 20 mM Tris with 50 mM sodium chloride pH 7.3, Pfs230D1M was added to the column in a five-fold molar excess and incubated overnight at 4 °C with rocking. Excess Pfs230D1M was washed out with 10 column volumes of wash buffer and HRV3C protease (Biovision part number 9206) was added at 0.125 times the mass of bound antibody and incubated overnight at 4 °C. After washing with five column volumes to recover protease and Fab, protease was removed by passing through a nickel column in the presence of 20 mM imidazole. The Pfs230D1M/Fab complex remained in the flow through and was concentrated and loaded onto a Hi-load 16/60 prep grade Superdex 75 size exclusion column (GE part number 17-1068-01) and fractions were recovered, and the complex was identified by SDS-PAGE. Pooled fractions were concentrated using Amicon Ultra 0.5 mL centrifugal filters part number UFC500396.

**Crystallization and structure determination.** Diffraction quality crystals were obtained using the hanging drop vapor diffusion method by mixing 1 μL of 3.6 mg/ml Pfs230D1M-4F12 Fab complex and 1 μL of precipitating solution that contained 100 mM sodium acetate (pH 4.6), 10–20% polyethylene glycol 4000, and 20% glycerol after extensive crystallization trials with various commercial crystallization screens setup with and without additives. The crystals were flash frozen in liquid nitrogen and data were collected at beamline 22-BM (SER-CAT) of the Advanced Photon Source at the Argonne National Laboratory on a MAR MX225 CCD detector. A crystal that diffracted to 2.38 Å resolution with cell dimensions (in Å) of $a = 79.95$, $b = 191.33$, and $c = 41.20$ and belonged to the space group $P2_12_12$ (Supplementary Table 1) was used to collect a native dataset. Multiple soakings of crystals with several heavy atoms did not produce a usable heavy atom derivative. Five X-ray datasets were processed, reduced and scaled with XDS[52]. The crystals were prone to radiation damage as data collection progressed. The earliest collected data were used for refinement, while maintaining a data completeness of greater than 95% and a multiplicity of measurement greater than 4. The structure of the complex was determined by molecular replacement using Phaser[53] and employed combinations of variable and constant domains from Fabs in our in-house data base as search models. The model of Pfs230D1 was constructed by iterative manual tracing of the chain using the program O[54] after each cycle of refinement with stepwise increase in the resolution using CNS[55]. The final cycles of refinement and model building were performed using PHENIX[56] and Coot[57], respectively. All structural figures were produced with the PyMOL molecular graphics system, version 1.7.4 (Schrödinger, LLC). Coordinates and structure factors were deposited in the Protein Data Bank (PDB) at the Research Collaboratory for Structural Biology (RCSB) with the identifying PDB code: 6OHG.

**Candidate variant dataset for analysis of amino-acid polymorphism within Pfs230D1M boundaries.** An initial list of candidate variants (polymorphisms) was obtained from the MalariaGEN Pf3k project pilot release 5 (MalariaGEN Pf3k Team 2016), which is a comprehensive collection of genome variation from thousands of *P. falciparum* samples collected from 14 countries. The project is run

by the Broad Institute, University of Oxford, and Wellcome Trust Sanger Institute. We downloaded a set of de novo genotype calls (In VCF format) for chromosome 2 (Pf3D7_02_v3) called on 2,640 samples using the *GATK3* best practices[58] and SnpEff[59] pipelines. Next, the *Bcftools ver. 1.9* package (http://github.com/samtools/bcftools) was used to subset only the calls between nucleotide positions 372,061–372,645 on Pf3D7_02_v3 that coincide with Pfs230D1. This resulted in an initial list of 36 candidate variants before ten failing GATK's quality filter (i.e., had a Low_VQSLOD score) and ten predicted to cause silent mutations were removed. Of the remaining 16 candidates, a variant (G > A) at Pf3D7_02_v3 position 372,250 was identified corresponding to a G605S missense mutation and with an AF across all samples of 0.94 (suggesting A rather than G is the major allele). Checks within individual populations showed A to be the major allele in all cases. To be consistent with the other polymorphisms (all minor alleles), we instead use the reference (G605) to represent this variant. Two different missense variants were called at Pf3D7_02_v3 position 372,574 (G > T, AF = 3.8E-4 and G > A, AF = 1.3E-3). As both are predicted to cause the same amino-acid change (D713Y), we display only 15 variants on the structure in Fig. 2c and list the higher AF of these two.

**Pearson correlation and t-SNE analysis of amino-acid polymorphism within Pfs230D1M boundaries**. To facilitate analysis of how the distribution of minor alleles across the 2640 *P. falciparum* samples relate between each pair of variants on Pfs230D1, candidate variant calls were converted into a binary format (0 for reference, 1, for *any* alternative allele present to collapse multi-allelic sites) to form a matrix. Any samples with missing or ambiguous calls were removed, and only variants passing *GATK's* quality filter and predicted to cause missense mutations were retained. This left a final binary matrix with 15 candidate missense variants as rows and 2618 remaining samples as columns.

For each pair of candidate variants, a Pearson correlation coefficient was calculated using R ver. 3.5.1[60]. Excluding self-correlations, Pearson's *r* values ranged from −0.013 to 0.666 and are shown in the left panel of Supplementary Fig. 3 as a hierarchical clustered heatmap generated with the *heatmaply* ver. 0.16 package in R[61]. A cluster of six mutations with the highest scoring correlations (P550S, K644Q, K661T, K665Q, D714N, and N732S) is shown in aqua green in the bottom left of the heatmap. K661N and T652R are also shown to cluster at the top right of the heatmap. *P*-values were calculated for each of the above tests and corrected for multiple-testing false discovery rate[62]. All pairwise correlations in the clusters mentioned above were highly significant ($p < 0.001$). While G605 showed the only two negative pairwise correlations ($r = 0.013$ for T652R and K661N), neither of these tests were found to be significant.

For a different visual assessment of cluster similarity across samples shown in Supplementary Fig. 3B, we used the t-Distributed Stochastic Neighbor Embedding (t-SNE) method[63] to project clusters in a two-dimensional space using R and the *Rtsne* package ver. 0.15 (default parameters plus: *perplexity = 4, theta = 0*)[64]. In brief, the method captures the distribution of points for "neighborhoods" of points in the original high-dimensional space (2618 for each sample in our case). It then tries to preserve these relationships when reconstructing the points in a desired lower-dimensional space. The variants that are closer in proximity on this plot have allele distributions across samples that are more similar than those drawn farther away on the plot.

**Culturing *P. falciparum* NF54 and NF54 Pfs48/45KO parasites**. In this study, the *P. falciparum* NF54 strain and *P. falciparum* NF54 Pfs48/45KO strain were used. The Pfs48/45KO strain was obtained from Dr. Robert Sauerwein. *P. falciparum* NF54 in vitro gametocyte cultures were maintained using methods described in *Methods in Malaria Research* 5th edition[65]. The parasites were cultivated in vitro in RPMI 1640 with human O⁺ erythrocytes and heat-inactivated serum 10% v/v, 0.0628 M sodium bicarbonate, and hematocrit was maintained at 5%, and cultures were gassed using a 5% $O_2$, 5% $CO_2$, and 90% $N_2$ gas mixture at 37 °C. While the Pfs48/45KO strain was maintained under similar conditions, pyrimethamine was added to a final concentration of 10 μM in the flask. The pyrimethamine pressure was maintained for three cycles, and then taken off pyrimethamine and used within two cycles. Gametocyte emergence was induced by a temperature change from 37 °C to room temperature.

**Live gamete labeling and analysis by confocal microscopy**. For macrogamete studies NF54 and NF54 Pfs48/45KO Day 14–16 cultures were enriched using the VarioMACS™ Separator in combination with CS Columns (Miltenyi Biotec) and 22 G flow resistor at room temperature. The CS column was placed in the Separator and equilibrated with 10 column volumes of RPMI 1640 with 0.5% bovine serum albumin added. Next, at most 50 mL of culture was passed through a column, which corresponds to two T75 tissue culture flasks. Subsequently, the column was washed with five column volumes of RPMI 1640 without bovine serum albumin. The column was removed from the magnetic field and then the cells were eluted with 30 mL of room temperature RPMI 1640. The gametes were allowed to emerge for an additional 15–20 min at room temperature. The enriched parasites were pelleted at $600 \times g$ for 8 min, the media was aspirated, and the pellet was resuspended in 1X PBS and then distributed between various mAb and PAb combinations. Purified mAb and PAb were used a concentration of 50 μg/mL,

diluted in 10% FBS in PBS, and incubated for 30 min at room temperature. The parasites were washed 3X in 1X PBS (Gibco). In certain cases, the parasites were pelleted and resuspended 1:1 in fetal bovine serum and smeared and then incubated with 2 μg/mL highly cross-absorbed, species-specific secondary antibodies for 30 min (goat α-rat, goat α-mouse or goat α-rabbit or goat α-human IgG) coupled with Alexa Fluor 488 (ThermoFisher Scientific) or Alexa Fluor 594 (ThermoFisher Scientific) along with Hoechst 33342 (ThermoFisher Scientific). The parasites were given another three washes in 1X PBS.

In cases where we were concerned in maintaining membrane integrity, such as in the Z-stack experiments, the parasites were plated onto Lab-Tek II sterile chambered Poly-L-Lysine coated #1.5 borosilicate welled cover-glass (Thermo Fisher Scientific). The slides were prepared 48 h ahead of the experiment, the sterile slides were coated with about 500 μL of 0.01% 70,000–150,000 mol wt Poly-L-Lysine (Sigma–Aldrich CAS:25988-63-0) and incubated for over two hours at room temperature. Next, the solution was aspirated, and the cover-glass was rinsed with 1X PBS and allowed to air-dry overnight to be used the next day.

For microgamete studies the NF54 SH and NF54 Pfs48/45KO Day 14–16 cultures were immediately spun down at $600 \times g$ for 6 min at 37 °C in a swinging bucket centrifuge. The media was aspirated, and the pellet was resuspended in warmed 1X PBS. The parasites were then incubated with various mAb and pAb combinations for 15–20 min. The parasites were then washed three times in 1X PBS and immediately resuspended 1:1 in fetal bovine serum and smeared.

**Fixed gamete labeling and analysis**. *Plasmodium falciparum* NF54 SH parasites were enriched using the method described above for macrogamete studies, and for microgamete experiments the parasites were prepared as described above. However, the parasites were not labeled with antibodies they were simply washed in 1X PBS thrice, pelleted and then resuspended 1:1 in fetal bovine serum. The slides were air dried and stored −70 °C or colder in heat-sealed packets with Drierite™ indicating desiccant (Sigma–Aldrich) prior to use. The slides were thawed for at least 30 min prior to immersion in cold methanol for 1 min and then air dried. Primary antibodies were diluted in 10% fetal bovine serum in PBS. Slides were labeled in a humidity box for 30 min or longer and washed thrice in PBS and then appropriately diluted species-specific antibodies were incubated for an additional 30 min or longer. The parasites were then washed again 3X in 1X PBS and then incubated with 1 μg/mL of DAPI. Confocal images were collected using a Leica SP8 confocal microscope equipped with a ×63/1.4NA oil-immersion objective, HyD detectors, an Argon 488 nm laser, a HeNe 594 nm laser, and a 405 nm diode laser. Quantification of double and single stained parasites was conducted using spot counting in Imaris software (Bitplane).

**Standard membrane feeding assays**. SMFAs were performed as previously described[16] using Protein G purified polyclonal antibodies (IgG) (pAbs) or mAbs dialyzed into PBS. Washed human red cells from freshly drawn blood and cultures containing mature gametocytes were resuspended to 60% hematocrit in native normal human serum (in the presence of complement). Red cells (120 μl) were mixed with 60 μl of appropriate dilutions of antibodies and 80 μl of parasites and fed to 50 female *A. freeborni* (NIH strain) with a feeding apparatus (a Baudruche membrane covering a glass water jacket heated at 40 °C). Mosquitoes were allowed to engorge for 15 min. The fed mosquitoes were maintained at 26 °C and 80% relative humidity. Seven to 8 days after feeding, the mosquitoes were dissected and their midguts examined for oocysts (products of parasite fertilization and midgut infection).

**Immunoassays (ELISAs, competition ELISAs, and western blotting)**. ELISAs and western blots for mapping the new Pfs230D1 mAb panel were performed essentially as previously described using the following reagents: anti-mouse IgG (gamma) antibody, human serum adsorbed and peroxidase-labeled (Seracare), and anti-mouse (H + L) antibody, rabbit and human serum adsorbed and phosphatase labeled (Seracare), respectively[66], using purified Pfs230NS or Pfs230D1M coating at a concentration of 1 μg/mL, 100 μL per well.

To evaluate if the conformational epitopes for mAbs 4F12 and 5H1 are located on different portions of the Pfs230D1 molecule, competition ELISA using biotinylated antigen was performed as previously reported[67] with minor modifications. Briefly, Pfs230D1M was biotinylated by using EZ-Link™ Sulfo-NHS-biotinylation kit (ThermoFisher Scientific, MA) following manufacturer's instructions. Standard ELISA plates were coated with one of the conformational mAb (in carbonate buffer, pH 9.5) and incubated overnight at 4 °C. All remaining steps were performed at room temperature unless otherwise specified, using Blocker BSA (10%) in TBS (Thermo Scientific) as BSA substitute for blocking and antigen/mAb dilutions as per manufacturer's recommendations; washing steps were performed in quadruplicate using TBS, 0.05% Tween 20, pH 7.4. After blocking (for 1 h at 37 °C) and washing, different amounts of competitor mAb (ranged from 0.16 to 2500 ng) were added per well (in triplicate) followed by 10 ng/well of biotinylated antigen. After 1 h incubation and corresponding washing, residual biotinylated Pfs230D1 was detected by using NeutrAvidin-HRP (Thermo Scientific) at 10 ng/well and 1-Step Ultra TMB-ELISA (Thermo Scientific) as developing solution, measuring the final absorbance at 450 nm.

**Isothermal calorimetry**. Pfs230D1M, Pfs230D1A, and 4F12 solutions were prepared for isothermal calorimetry (ITC) experiments by dialysis against PBS for 2 h. Prior to each experiment, the Pfs230D1M, Pfs230D1A, and 4F12 protein solutions were filtered through a Whatman inorganic membrane filter or 0.22 μm filter, checked for the presence of aggregates by dynamic light scattering using Malvern Nano-S instrument, and degassed for 30 min using the TA Instruments degassing station. The solutions containing Pfs230D1M or Pfs230D1A were loaded in the syringe at 165 μM or 49 μM, and 4F12 was placed in the calorimeter cell at 23 μM or 3 μM, respectively. Binding experiments were performed on a TA Instruments Low Volume Nano ITC instrument at 25 °C using 20 injections of 2.5 μL each, with injection interval of 300 s, and a syringe stirring speed of 250 rpm. The initial injections were excluded from the data analyses. The ITC data were fitted to the independent binding site model using the TA Instruments NanoAnalyze software.

**Statistics and reproducibility**. All studies were performed a minimum of two times as biological replicates. Three technical replicates were performed for each biological replicate when suitable e.g., ELISAs. The sample sizes used in the ex vivo SMFA are noted in the tables or figures. All individual SMFA assessments included ≥20 mosquitoes condition tested. Zero-inflated negative binomial models were fitted to oocyst counts and parameterized to test the difference among mAb groups with both epitopes versus groups with the single epitope. Confidence intervals for TR activity were calculated from these models and then TBA and corresponding confidence intervals were calculated using a method described previously[68].

**Reporting summary**. Further information on research design is available in the Nature Research Reporting Summary linked to this article.

## Data availability
The DNA sequences for the 4F12 heavy and light chains were submitted to the National Center for Biotechnology Information. The accession numbers for the 4F12 H and L chain are MK756320 and MK756321, respectively. Coordinates and structure factors were deposited in the PDB at the Research Collaboratory for Structural Biology (RCSB) with the identifying code: 6OHG.

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

## Acknowledgements

We thank Marian Ortiz-Rodriquez and Deepika Seethamraju for technical assistance with the production of the *P. pastoris* expressed Pfs48/45 and Pfs230D1A, respectively. We appreciate Dr. Robert Sauerwein for providing the Pfs48/45KO parasite line for confocal imaging studies, and J. Patrick Gorres for assisting in editing and preparation of this manuscript. X-ray data were collected at Southeast Regional Collaborative Access Team (SER-CAT) 22-BM beamline at the Advanced Photon Source, Argonne National Laboratory. SER-CAT is supported by its member institutions (see www.ser-cat.org/members.html), and equipment grants (S10_RR25528 and S10_RR028976) from the National Institutes of Health. Use of the Advanced Photon Source was supported by the U. S. Department of Energy, Office of Science, Office of Basic Energy Sciences, under Contract No. W-31-109-Eng-38. This research was supported by the Intramural Research Program of NIAID, NIH.

## Author contributions

Designed research: K.S., A.G.G., D.V., D.N.G., and D.L.N.; Performed research: K.S., M.B., S.N., R.H., O.M., A.G.G., E.K., K.R., M.S., D.V., R.S., V.N., B.Z., and N.J.M.; Contributed unique reagents: B.Z.; Analyzed data: K.S., A.G.G., M.S., D.V., B.J.S., P.E.D., D.N.G., and D.L.N.; Wrote the paper: D.N.G. and D.L.N. and all other authors contributed to writing the manuscript.

## Competing interests

The authors declare no competing interests.
