## [Peer Review File · Communications Biology]

Reviewers' comments:

Reviewer #1 (Remarks to the Author):

The manuscript by Singh et al. is clear and well written. With the aim to advance the development of a *P. falciparum* transmission blocking vaccine, the authors have determined the crystal structure of Pfs230D1 in complex with the Fab fragment of inhibitory mAb 4F12. They have further studied the cellular localization of Pfs230 on the surface of sexual stages parasites and the effect of combining mAbs against Pfs230 and Pfs48/45.

In general, the study is interesting and clearly important.

Specific comments:

Line 116 and forward. The authors determined the crystal structure of an antigen-antibody complex. Please discuss to what extent antibody binding modified the structure of Pfs230D1.

Line145. Curiously, mAb4F12 seems to recognize amino acid sequences spanning the N585 position, which was mutated to glutamine in Pfs230D1. Apparently, this mutation has little impact on antibody binding. The manuscript could benefit from a more thorough characterization / discussion of the B-cell epitope recognized by mAb4F12. Which positions are critical for binding etc. Also, please visualize how the cysteine connectivity affects binding.

Line 269. It has been the general understanding that Pfs230 binds to the parasite surface through interactions with Pfs48/45. Your data seems to indicate that this interaction is not always required for surface exposure. Could you please elaborate on this phenomenon? F.x. does the new crystal structure together with published structures of 6-cys domains help to understand the interaction of Pfs230 with protein domains from other parasite proteins including Pfs48/45?

Line 256. This synergy is clearly of interest. Please discuss it in the context of the cellular localization of Pfs230 and the observation that surface exposure of Pfs230 is not always dependent on Pfs48/45 expression.

Tables and figures are appropriate

Reviewer #2 (Remarks to the Author):

This manuscript describes the structural and functional characterisation of the important gamete surface protein and transmission blocking vaccine candidate Pfs230. It is an interesting manuscript and contains very useful information for the malaria vaccine development community. Major findings include the first structure of the D1 domain of Pfs230, which is in clinical trials as a malaria vaccine, and the identification and characterisation of some novel antibodies against this domain. There are, however, a few issues as outlined below. In particular, I was not convinced that the data supported claims in lines 33-37 of the abstract and more work is required to refine the structure to high quality. Nevertheless, I would support the publication of most of the data in this manuscript, with careful reinterpretation of some of the experiments.

Major comments:

The authors begin by cloning the 4F12 antibody and validating its effect on transmission. This study appears to show that antibody 5H1 is more effective in these assays than 4F12 but the authors do not comment on this at this stage? They then proceed to determine the structure of the 4F12 Fab fragment in complex with the Pfs230D1M domain. This structure is largely convincing, although the refinement statistics are rather poor for a structure at 2.38Å resolution. RSRZ outliers, side chain outliers and Ramachandran outliers are all very high for this resolution. More time is required on refinement to fix these issues with the model and this will give a clear view of the binding epitope for this antibody.

The authors make an interesting observation about the role of the N585 residue and its role in binding. They have mutated this residue to Q to avoid glycosylation and they then discover that this residue is part of the binding site. They state that it makes 13 contacts with 4F12 (line 148) which seems very unlikely for a single medium sized residue. It would be good to check this. They then measure the affinity and I presume that this is for the N585Q mutant, although this is not stated. However, they have not compared this affinity with that of the N585 wild type. This is an important control and the protein could be obtained by expression of wild-type followed by enzymatic deglycosylation. Alternatively, the affinity of the N585A mutant could show if this residue is important for binding.

The authors have a nice discussion of the similarity of their structure to other 6-cys domains. In their section in 166-173 they discuss the conformation of the loops and they might like to speculate about whether these loops might adopt different conformations when in the context of Pfs230. They might be able to speculate from their position relative to structurally rigid features of the domain

The discussion of the degree of polymorphism is nicely done and of significant value but the authors should consider a better way to visualise this. Perhaps different colours for residues polymorphic to different degrees on a surface representation of the domain, with an indication of the epitope might give a better at-a-glance view of this.

The discovery of four new mAbs, and the identification of 5H1 as a mAb with affinity better than that of 4F12 is an interesting development. The analysis of the binding of 5H1 and 4F12 to macro- and microgametes is also very interesting, suggesting different targets for these two antibodies.

The section in lines 234-255 is troubling for a couple of reasons. Firstly, the mapping of antibody 3E12 to the C-terminal domain of Pfs48/45 is not convincing as the quality of the protein used does not look adequate from Figure S5. It is very unlikely that 6-cys domains produced in *E. coli* are correctly folded and no evidence is presented to show that they are in this case. I would therefore not be convinced that this recognition is specific. Secondly, the finding that 3E12 staining does not overlap with Pfs230 staining (done with polyclonal rather than 4F12?) is over interpreted as proving that Pfs230 can be membrane associated without Pfs48/45. Perhaps the presence of Pfs230 covers up the Pfs48/45 epitope in some situations? To help with this the authors have studied Pfs48/45 KO parasite lines, and see Pfs230 on the surface, which is a sensible experiment. However, as far as I can tell, they use 4F12 for this experiment and polyclonal for the previous experiment, making the two non-comparable. If these two pieces of data are to be compared to draw a conclusion, the same antibodies should be used for both. Otherwise, what if one shows cross-reactivity and one doesn't? Or the epitopes are different for the two and so steric effects differ?

Finally, a nice set of data is presented on considering the effect of comparing the activities of 5H1 and 4F12. The finding that these two antibodies do not compete for binding appears convincing, as does the demonstration of synergy in transmission blocking activity.

Minor comments.

Line 115 Would be good to have a figure to show the transmission blocking activity as well as a table.

Line 290-291 This section lacks supporting data. Presumably a few structure-guided sequence alignments could easily show if this is the case or not?

General reviewer summations:

Italics note reviewers' comments:

In particular, the refinement of the structural model requires more work to improve the statistics and present a better model.

Please see our response to Reviewer #2 on our efforts to improve the statistics of the model.

In addition, a comparison of the affinities of N585 wildtype and the N585Q mutant,

Please see our response to Reviewer #1.

evidence for the correct folding of the 6-cys domains produced in E. coli,

Please see our response to Reviewer #2

and the consistent use of the 4F12 antibody for the experiments with the Pfs48/45 KO parasite lines would be beneficial for the manuscript.

Please see our response to Reviewer #2

Mandatory checklists are completed:

- 1. Editorial policy checklist*
- 2. Reporting summary*

Reviewer #1 (Remarks to the Author):

The manuscript by Singh et al. is clear and well written.

We thank the reviewer for this feedback.

With the aim to advance the development of a P. falciparum transmission blocking vaccine, the authors have determined the crystal structure of Pfs230D1 in complex with the Fab fragment of inhibitory mAb 4F12. They have further studied the cellular localization of Pfs230 on the surface of sexual stages parasites and the effect of combining mAbs against Pfs230 and Pfs48/45.

In general, the study is interesting and clearly important.

Specific comments:

Line 116 and forward. The authors determined the crystal structure of an antigen-antibody complex. Please discuss to what extent antibody binding modified the structure of Pfs230D1.

We did not detect alterations in the Pfs230D1 structure due to antibody binding. To properly study any differences, we would need the structure of Pfs230D1 by itself, not bound to an antibody. Using the Pfs230D1M protein by itself, we performed exhaustive crystallization experiments, but crystals did not appear.

Line 145. Curiously, mAb 4F12 seems to recognize amino acid sequences spanning the N585 position, which was mutated to glutamine in Pfs230D1(M). Apparently, this mutation has little impact on antibody binding.

Yes, the mAb 4F12 recognizes the new glutamine (Q) at position 585 that was introduced to remove the glycosylation site, NNT at positions 585-587. We produced a new Pfs230D1 construct (Pfs230D1A), that inactivates the N-linked site with a T587A mutation and reverts position 585 to the wild-type N. T587 is outside the edge of the binding epitope and does not participate in binding. Isothermal titration calorimetry shows that the K_d (45nM) using the new construct is within 2-fold of the K_D (24nM) using Pfs230D1M (N585Q) (Fig. S1). Our attempts to determine a structure of the new Pfs230D1A with the Fab has not yielded crystals.

The manuscript could benefit from a more thorough characterization / discussion of the B-cell epitope recognized by mAb4F12. Which positions are critical for binding etc.

Based on the structure and including all atoms of Pfs230D1 that are within 4 angstroms of one or more atoms of the 4F12 Fab, the B-cell epitope on Pfs230D1 consists of atoms from these residues: K581, Y582, A583, S584, Q585, N586, D594, T596, D597, Q598, K600, P601, T602, E603, S604, K607, K609. We do not know which residues are critical for binding, except of course that we know that N585 can be a Q and that T587 can be an A, without affecting the binding affinity more than 2-fold.

New sentences in text:

"There are light chain contacts to the loop between the second and third beta-strands at T596, D597, Q598, K600, T602, E603, S604, and K607. The light chain also contacts the third beta-strand at K609. The first disulfide (593-611) links the two beta-sheets at the second and third strands and appears to stabilize the 4F12 epitope on Pfs230D1. The 4F12 heavy chain binds Pfs230D1 at the loop between the second and third beta strands of Pfs230D1, contacting P601 and overlapping the light chain contacts to atoms of D597 and K600."

Also, please visualize how the cysteine connectivity affects binding.

We thank the reviewer for this suggestion. The first disulfide (593-611) is between the two beta-sheets and links the second and third strands, which are part of the epitope.

This sentence has been added to the text:

"The first disulfide (593-611) links the two beta-sheets at the second and third strands and appears to stabilize the 4F12 epitope on Pfs230D1."

Line 269. It has been the general understanding that Pfs230 binds to the parasite surface through interactions with Pfs48/45. Your data seems to indicate that this interaction is not

always required for surface exposure. Could you please elaborate on this phenomenon? F.x. does the new crystal structure together with published structures of 6-cys domains help to understand the interaction of Pfs230 with protein domains from other parasite proteins including Pfs48/45?

We strove to glean information about the interactions of 6-cys domains from our Pfs230D1 structure. We suggest that more structures of complexes and multi-domain proteins like Pfs230 are needed.

Line 256. This synergy is clearly of interest. Please discuss it in the context of the cellular localization of Pfs230 and the observation that surface exposure of Pfs230 is not always dependent on Pfs48/45 expression.

As part of the discussion, the following sentences have been added:

"Understanding the domain packing of Pfs230D1 based on epitope exposure appears even more complicated given the observation that Pfs230 may be membrane associated in the absence of its known binding partner Pfs48/45, a GPI-anchored protein (22,23). The biological basis for the membrane association of Pfs230 in the absence of Pfs48/45 remains unclear. Pfs230 is a large 230 kDa protein with fourteen 6-cysteine rich domains. It remains to be determined if other domain(s) contribute to its membrane association or to its potential to bind to other membrane associated proteins. We have not identified any novel sexual stage proteins following immunoprecipitation of Pfs230 from solubilized NF54 sexual stage parasites other than Pfs48/45 by tryptic digestion followed by liquid chromatography with tandem mass spectrometry. Additional work is warranted using the Pfs48/45 KO parasite line."

Tables and figures are appropriate
We appreciate the reviewer's comment.

Reviewer #2 (Remarks to the Author):

This manuscript describes the structural and functional characterisation of the important gamete surface protein and transmission blocking vaccine candidate Pfs230. It is an interesting manuscript and contains very useful information for the malaria vaccine development community. Major findings include the first structure of the D1 domain of Pfs230, which is in clinical trials as a malaria vaccine, and the identification and characterisation of some novel antibodies against this domain. There are, however, a few issues as outlined below. In particular, I was not convinced that the data supported claims in lines 33-37 of the abstract and more work is required to refine the structure to high quality.

Please see comments below to address the concerns.

Nevertheless, I would support the publication of most of the data in this manuscript, with careful reinterpretation of some of the experiments.

We appreciate this support.

Major comments:

The authors begin by cloning the 4F12 antibody and validating its effect on transmission. This study appears to show that antibody 5H1 is more effective in these assays than 4F12 but the authors do not comment on this at this stage?

We have cloned H and L chains from the 5H1 cell line but the recombinantly expressed IgG does not bind Pfs230D1M by ELISA or Western blot. We are continuing our efforts to clone this mAb. We inserted the following sentence to explain why we were unable to proceed with 5H1.

Line 227. Inserted the following sentence: "It has not yet been possible to clone and express a recombinant form of 5H1."

They then proceed to determine the structure of the 4F12 Fab fragment in complex with the Pfs230D1M domain. This structure is largely convincing, although the refinement statistics are rather poor for a structure at 2.38Å resolution. RSRZ outliers, side chain outliers and Ramachandran outliers are all very high for this resolution. More time is required on refinement to fix these issues with the model and this will give a clear view of the binding epitope for this antibody.

We strenuously tried to improve the outlier stats on this model, but at the end, were limited by the quality of X-ray data obtainable from these crystals.

We realized that the percent of outliers was high. We exhaustively refined the model against our top five X-ray datasets, aiming to improve the RSRZ, side chain, and Ramachandran statistics. The final dataset and model represented the best of the five. Overall, we collected data on 35 crystals with resolutions between 2.38 and 2.9. While studying the datasets, we found that these small crystals suffered radiation damage within the first 50-60 degrees of data collection. As radiation damage proceeds, the high-resolution reflections begin to fade and the Rmerge at high resolution begins to climb. We reintegrated the early data frames, choosing a sweep of X-ray data that still resulted in a minimum of 95% completeness and multiplicity greater than 4.

Five datasets were used for refinement. The electron density showed that some of the Pfs230D1M molecules showed breakage of the 593-611 disulfide. This led to small movements locally in those molecules, leading to heterogeneity, which is difficult to model. As a result, the statistics of the structure suffered. We thought to reduce the resolution of the data used for refinement but found that the addition of data to 2.38Å improved the electron density markedly in some areas of the maps. We submit that our delineation of the binding epitope would not change significantly, even if improved statistics could be obtained.

We have added these lines to the Methods section:

"Five X-ray datasets were processed, reduced and scaled with XDS (52). The crystals were prone to radiation damage as data collection progressed. The earliest collected data were used for refinement, while maintaining a data completeness of greater than 95% and a multiplicity of measurement greater than 4."

The authors make an interesting observation about the role of the N585 residue and its role in binding. They have mutated this residue to Q to avoid glycosylation and they then discover that this residue is part of the binding site. They state that it makes 13 contacts with 4F12 (line 148) which seems very unlikely for a single medium sized residue. It would be good to check this.

We were surprised that five atoms of Q585 make thirteen 4-angstrom contacts to eight atoms of the antibody light chain. Below is a bit of the output from the CCP4 program NCONT, where the /A/ chain is Pfs230 and /B/ is the antibody light chain:

```
1/1/A/ 585 (GLN) . / CB [ C] :/1/B/ 28 (SER) . / O [ O] : 3.74
2                                     /1/B/ 92 (TYR) . / OH [ O] : 3.70
3/1/A/ 585 (GLN) . / CG [ C] :/1/B/ 27 (GLN) . / CB [ C] : 3.96
4                                     /1/B/ 27 (GLN) . / OE1 [ O] : 3.77
5                                     /1/B/ 28 (SER) . / O [ O] : 3.78
6/1/A/ 585 (GLN) . / CD [ C] :/1/B/ 92 (TYR) . / CE1 [ C] : 3.59
7                                     /1/B/ 92 (TYR) . / OH [ O] : 3.75
8/1/A/ 585 (GLN) . / OE1 [ O] :/1/B/ 2 (ILE) . / CD1 [ C] : 3.44
9                                     /1/B/ 92 (TYR) . / CD1 [ C] : 3.89
10                                    /1/B/ 27 (GLN) . / NE2 [ N] : 3.98
11                                    /1/B/ 92 (TYR) . / CE1 [ C] : 3.27
12/1/A/ 585 (GLN) . / NE2 [ N] :/1/B/92 (TYR) . / CE1 [ C] : 3.75
13                                    /1/B/ 92 (TYR) . / OH [ O] : 3.49
```

They then measure the affinity and I presume that this is for the N585Q mutant, although this is not stated. However, they have not compared this affinity with that of the N585 wild type. This is an important control and the protein could be obtained by expression of wild-type followed by enzymatic deglycosylation. Alternatively, the affinity of the N585A mutant could show if this residue is important for binding.

We describe a new construct with N at 585 and A at 587 (Pfs230D1A) above in this point-by-point document. It has a similar affinity as does the N585Q mutant (Pfs230D1M).

The authors have a nice discussion of the similarity of their structure to other 6-cys domains. In their section in 166-173 they discuss the conformation of the loops and they might like to speculate about whether these loops might adopt different conformations when in the context of Pfs230. They might be able to speculate from their position relative to structurally rigid features of the domain.

We speculate in the discussion that "These observed differences in the loops of the known domains may stem from the requirements of packing 6-cysteine domains in multi-domain proteins." We meant, a bit cryptically, that long loops are flexible and shorter loops are less flexible. Both kinds of loops are likely to be involved in the packing of 6-cys domains in Pfs230.

The discussion of the degree of polymorphism is nicely done and of significant value but the authors should consider a better way to visualise this. Perhaps different colours for residues

polymorphic to different degrees on a surface representation of the domain, with an indication of the epitope might give a better at-a-glance view of this.

We have added a panel "D" to Fig. 2, that shows the surface representation of Pfs230D1 in complex with a ribbon portrayal of the 4F12 Fab, with the polymorphic sites identified. We describe in the new Fig. 2 legend that 4F12 binds to a non-polymorphic surface of Pfs230D1.

The discovery of four new mAbs, and the identification of 5H1 as a mAb with affinity better than that of 4F12 is an interesting development. The analysis of the binding of 5H1 and 4F12 to macro- and microgametes is also very interesting, suggesting different targets for these two antibodies.

We appreciate the comment.

The section in lines 234-255 is troubling for a couple of reasons. Firstly, the mapping of antibody 3E12 to the C-terminal domain of Pfs48/45 is not convincing as the quality of the protein used does not look adequate from Figure S5. It is very unlikely that 6-cys domains produced in E. coli are correctly folded and no evidence is presented to show that they are in this case. I would therefore not be convinced that this recognition is specific.

We understand the reviewer's concerns. Our initial assessment of the epitope identification of 3E12 used limited enzymatic digestion of native Pfs48/45 and Western blotting followed by in gel digestion coupled with MS/MS for fragment identification. We identified that it was the carboxyl-terminal region of Pfs48/45 that reacted with mAb 3E12. This led to the development of the recombinant expression of domain 3 which is presented in the *SI Appendix* Fig. S6 (revised numbering). Using this observation, a colleague has successfully generated a recombinant fusion protein using a mammalian expression system that is comprised of Pfs230D1 fused with Pfs48/45D3. The secreted recombinant fusion protein reacts by Western blot with Pfs230D1 mAbs (not shown) and mAb 3E12 in an oxidized state only. We provide this information and figure for review only as our colleague will be reporting this work in the near future.

Secondly, the finding that 3E12 staining does not overlap with Pfs230 staining (done with polyclonal rather than 4F12?) is over interpreted as proving that Pfs230 can be membrane associated without Pfs48/45. Perhaps the presence of Pfs230 covers up the Pfs48/45 epitope in some situations? To help with this the authors have studied Pfs48/45 KO parasite lines, and see Pfs230 on the surface, which is a sensible experiment. However, as far as I can tell, they use 4F12 for this experiment and polyclonal for the previous experiment, making the two non-comparable. If these two pieces of data are to be compared to draw a conclusion, the same antibodies should be used for both. Otherwise, what if one shows cross-reactivity and one doesn't? Or the epitopes are different for the two and so steric effects differ?

To address the reviewer's concern, we have performed an additional labeling study on the Pfs48/45KO parasite using the same series of reagents for Fig. 4 H and I. Similar observations were made regardless of the Pfs230 and Pfs48/45 specific reagents used in the studies. We have updated Fig. 4 to show the results with similar reagents and inserted the previous Fig. 4 Panel I in *SI Appendix* Fig. S6F for supporting information.

Finally, a nice set of data is presented on considering the effect of comparing the activities of 5H1 and 4F12. The finding that these two antibodies do not compete for binding appears convincing, as does the demonstration of synergy in transmission blocking activity.

We appreciate the reviewer's comments regarding the combination studies.

While preparing the revised manuscript, we completed additional combinatorial feeds using the chimeric human 4F12 in combination with 3E12 to demonstrate that transmission reducing activity is enhanced with complement activation as well as without. A figure showing this activity has been inserted in Fig. 4C and a replicate assay is included in *SI Appendix* Fig. S5C. The legends for each figure have been revised to include similar results as previously reported.

A sentence has been inserted in results, line 274, "We assessed the activity of both 4F12 or rh4F12 in combination with 5H1 at concentrations below those that completely block transmission, and found that the combinations demonstrated enhanced TR activity (Fig. 4B and C)."

Minor comments.

Line 115 Would be good to have a figure to show the transmission blocking activity as well as a table.

As the results of transmission blocking shown in the table represents 8 independent studies, we feel the data are better conveyed in a table. A graphical presentation would require some form of normalization which would obscure the inherent variation between each experimental data set and in particular the oocyst densities of each control group.

Line 290-291 This section lacks supporting data. Presumably a few structure-guided sequence alignments could easily show if this is the case or not?

The section to which the reviewer is referring is in the Discussion:
Lines 290-291 (currently lines 302-303):

"The six 6-cysteine domains with available structures overlay structurally with an rmsd of about 2Å. We observed that one end of all six domains has short loops and the other end of each domain has long loops. We hypothesize that the other thirteen domains of Pfs230 and of other 6-cysteine family members will also have such loop characteristics."

As all six crystallographically-determined 6-cysteine domain structures exhibit the "short loops-long loops phenotype", we predict that the other 6-cysteine domains will too. We understand that the reviewer would like us to check the predictions of our hypothesis by modeling or alignments

using the sequences of other 6-cys domains. Our predictions are based on six X-ray structures at the moment, which we count as strong evidence. We would prefer to allow other workers to extend the prediction by modeling other 6-cysteine domains.

REVIEWERS' COMMENTS:

Reviewer #1 (Remarks to the Author):

All queries and comments have been answered adequately.

Reviewer #2 (Remarks to the Author):

The authors have addressed the majority of my concerns and I am happy to recommend publication.

A few very small points:

Table 1 really needs the CC1/2 for the data set

On the issue of Q585 making 13 interactions, the chemistry shown in the table doesn't make sense to me. What chemical bonds are they suggesting the CB of a GLN makes with an O or OH? Also all of the bonds are $>3.2\text{\AA}$. Even those which are likely to be hydrogen bonds will therefore be very weak. I would drop the 13 bond claim, as I don't think that the data holds up.

I also found the response to the comment about lines 290-291 to be a little strange. A quick structure-based sequence alignment, taking half an hour, will show if this speculation is true or not. So why not either delete the point or confirm it if it is important?

Otherwise I found the responses convincing.

Please find our remarks for rebuttal to Reviewer #2:

Reviewer #2 (Remarks to the Author):

The authors have addressed the majority of my concerns and I am happy to recommend publication.

A few very small points:

Table 1 really needs the CC1/2 for the data set

We inserted the CC1/2 statistic for the data set into Supplementary Table 1.

On the issue of Q585 making 13 interactions, the chemistry shown in the table doesn't make sense to me. What chemical bonds are they suggesting the CB of a GLN makes with an O or OH? Also all of the bonds are $>3.2\text{\AA}$. Even those which are likely to be hydrogen bonds will therefore be very weak. I would drop the 13 bond claim, as I don't think that the data holds up.

We stated that there are 13 contacts, each less than 4 angstroms. The reviewer uses the words "interactions" and "bonds" to refer to what we term "contacts". Van der Waals forces, hydrophobic forces, and simple packing are some of the phenomena that can be involved between atoms of any nature. Much more biophysical analysis would be needed to characterize the forces, if any, between the 13 pairs of atoms. We thought it noteworthy that a mutation from N to Q within an antibody epitope could be accommodated without a detectable change in the binding affinity of the antibody. Thus, we reported our observation.

We have removed any reference to the thirteen contacts made by Q585.

I also found the response to the comment about lines 290-291 to be a little strange. A quick structure-based sequence alignment, taking half an hour, will show if this speculation is true or not. So why not either delete the point or confirm it if it is important?

We have deleted the point as the reviewer suggests.